# Effects of Supplementation of *Piper sarmentosum* Leaf Powder on Feed Efficiency, Rumen Ecology and Rumen Protozoal Concentration in Thai Native Beef Cattle

**DOI:** 10.3390/ani9040130

**Published:** 2019-03-29

**Authors:** Anusorn Cherdthong, Benjamad Khonkhaeng, Suban Foiklang, Metha Wanapat, Nirawan Gunun, Pongsatorn Gunun, Pin Chanjula, Sineenart Polyorach

**Affiliations:** 1Tropical Feed Resources Research and Development Center (TROFREC), Department of Animal Science, Faculty of Agriculture, Khon Kaen University, Khon Kaen 40002, Thailand; k.benjamad@hotmail.com (B.K.); metha@kku.ac.th (M.W.); 2Faculty of Animal Science and Technology, Maejo University, Chiangmai 50290, Thailand; bungung@hotmail.com; 3Program in Animal Production Technology, Faculty of Technology, Udon Thani Rajabhat University, Udon Thani 41000, Thailand; nirawan_kku@hotmail.com; 4Department of Animal Science, Faculty of Natural Resources, Rajamangala University of Technology-Isan, Sakon Nakhon Campus, Phangkhon 47160, Sakon Nakhon, Thailand; pongsatorng@hotmail.com; 5Department of Animal Science, Faculty of Natural Resources, Prince of Songkla University, Songkhla 90112, Thailand; pin.c@psu.ac.th; 6Department of Animal Production Technology and Fisheries, Faculty of Agricultural Technology, King Mongkut’s Institute of Technology Ladkrabang, Bangkok 10520, Thailand; neenart324@hotmail.com

**Keywords:** tropical plant, rumen ecology, methane, ruminant

## Abstract

**Simple Summary:**

This work was to study the influence of *Piper sarmentosum* (PS) leaf powder supplementation in cattle diet. It was found that supplementation of PS leaf powder at 2.4 g/d resulted in improving total feed intake and dry matter (DM) digestibility whereas there was reduced protozoal population and methane (CH_4_) production in cattle.

**Abstracts:**

The aim of this research was to study the influence of inclusion of *Piper sarmentosum* (PS) leaf powder on feed efficiency, rumen ecology, protozoal diversity and CH_4_ emission in cattle. Four male beef cattle (1–1.5 years old) with similar initial body weights (BW) of 150 ± 20.00 kg were randomly assigned to a 4 × 4 Latin squared design. Experimental diets consisted of four levels of PS leaf powder supplementation at 0 (control), 0.6, 1.2 and 2.4 g/head/d, respectively. Dry matter (DM) intake of rice straw and total intake in terms of g/kg BW^0.75^ differed among treatments and were linearly greatest when PS was added at 2.4 g/head/d (*p* < 0.05). PS leaf powder did not affect the intake of nutrients and digestibility of organic matter, crude protein, and fibers. Nevertheless, digestibility of DM was found to increase by 4.8% in cattle when added at 2.4 g/head/d as compared to the other control treatments. PS leaf powder did not affect ruminal pH, ruminal temperature, ammonia–nitrogen concentration and blood urea-nitrogen (*p* > 0.05). The bacterial population was similar across PS levels (*p* > 0.05). However, the protozoal count was lower in animals fed with the supplement at 2.4 g/head/d (*p* < 0.05). The PS leaf powder did not affect total volatile fatty acid (VFA) and acetic acid (C2), butyric acid (C4), C2 to propionic acid (C3) and C2 to C3 to C4. However, concentration of C3 at 4 h post feeding and mean value linearly increased with supplementation of PS leaf powder at 2.4 g/d. The 4 h post-feeding and mean values of CH_4_ concentration were linearly reduced with PS supplementation (*p* < 0.05). An increase of PS leaf powder at 2.4 g decreased CH_4_ after feeding 4 h by 21.33% when compared to no PS leaf powder. Thus, supplementation of PS leaf powder at 2.4 g in ruminant feeding is recommended for manipulating rumen efficiency.

## 1. Introduction

Methane (CH_4_) is an end-product of enteric fermentation in cattle [1]. Hydrogen is transferred via the action of a group of bacteria called methanogens, which contribute their energy by attaching carbon dioxide to hydrogen to form CH_4_ [2]. Thus, approaches to limiting CH_4_ emissions must include an alternative mechanism to eliminate hydrogen.

Various studies have demonstrated that the use of herbs or plant extracts could manipulate rumen ecology in terms of reducing CH_4_ emissions [3,4]. In many feed additives, anti-microbial activity is short-term as the rumen adjusts to neutralize the newly introduced chemical [5]. The most promising avenues for reducing CH_4_ emissions are the development of new products for decreasing protozoal counts in the rumen and the use of plant secondary compounds that specifically target methanogen [2]. Recently, tropical plant herbs have been widely used as anti-methanogenic bacteria and extensively used to balance the bacterial ecology of the rumen particularly in tropical areas [6].

*Piper sarmentosum* Roxb. (PS), which is widely abundant in tropical areas including Thailand, is often used as a food flavoring agent, in traditional medicine and for pest control [7]. PS species contain various bioactive compounds, such as alkaloids, amides, pyrones, dihydrochalcones, phenylpropanoids, lignans neolignans and flavonoids [8]. In addition, the negative effects of flavonoid-rich plants on CH_4_ emission and methanogenic bacteria in vitro and in vivo have been determined [9,10]. Supplementation of flavonoid substances could manipulate the rumen ecology of dairy cows by enhancing milk production [11], protecting ruminal acidosis [12], decreasing CH_4_ emission and lowering microbial populations such as protozoal population and methanogenic bacteria [13]. Furthermore, protozoa are significant factors in methanogenesis in the rumen due to methanogenic bacteria attached to their surfaces. Flavonoid-rich plants decrease the cilia-associated methanogenic bacteria and, thus, reduce CH_4_ emissions [14]. Patra and Saxena [9] revealed that flavonoid-rich plants provided direct effects that inhibit methanogenic bacteria and lowered protozoa related to rumen methanogenesis. Use of PS as a feed additive has been reported to improve the average daily gain, antioxidant capability and immune system of piglets [15]. However, there is no information on the effect of PS as a feed additive in ruminants. Thus, the aim of this research was to study the influence of inclusion of PS leaf powder on feed efficiency, rumen ecology, protozoal diversity and CH_4_ emission in cattle.

## 2. Materials and Methods

### 2.1. Animals and Dietary Treatments

Thai native beef cattle used in this experiment were endorsed by the Animal Ethics Committee of Khon Kaen University (approval number AEKKU 18/2558). Four male beef cattle (1–1.5 years old) with similar initial body weights (BW) of 150 ± 20.00 kg were randomly assigned to a 4 × 4 Latin squared design. Experimental diets consisted of four levels of *Piper sarmentosum* (PS) leaf powder supplementation at 0 (control), 0.6, 1.2 and 2.4 g/head/d, respectively. All PS used in the present experiment was harvested at the age of 60 days after planning in Khon Kaen province, Thailand. Fresh PS leaf meal were collected, and sun-dried for 2 to 3 days, then ground to pass through a 1-mm sieve before use as a supplement with the animals.

All beef cattle were kept in separate pens, and clean fresh water and mineral blocks were available at all times. Concentrates were fed at 0.5% BW daily, and rice straw was fed *ad libitum* at 07:00 and 16:00. PS was also supplemented twice daily during respective treatment periods. The study was consisted of four periods of 21 days each. During the first 14 days, all animals were fed their respective treatments with *ad libitum* intake, but during the last 7 days, they were moved to metabolism crates for total fecal collection to assess nutrient digestibility. The proportion of ingredients in concentrate and nutrient contents of concentrate, rice straw and PS powder are shown in Table 1.

### 2.2. Data and Sample Collection

Feed samples of roughage, concentrate and PS leaf meal were determined daily by weighing the offered and uneaten feed during the morning feeding. During the last 7 days of each period, samples of the concentrate mixture, roughage, PS leaf powder, refusals and feces were collected daily.

Fecal samples were collected during the last 7 days of each period using the total collection method, as the animals were in the metabolism crates to study nutrient digestibility. The fecal samples collected were about 5% of total fresh weight and divided into two parts; the first part was used for DM analysis every day, and the second part was kept in refrigerator and pooled by animal at the end of each period for chemical analysis. The samples were dried at 60 °C and ground (1 mm screen using a Cyclotech Mill, Tecator, Hoganas, Sweden) and analyzed for DM, nitrogen, organic matter (OM) [16], neutral detergent fiber (NDF) and acid detergent fiber (ADF) according by Van Soest et al. [17]. The total flavonoid concentration was analyzed using the Folin–Ciocalteu reagent method [18]. Absorbance was evaluated at 725 nm spectrophotometrically (Shimadzu, Kyoto, Japan).

At the end of each period, rumen fluid and jugular blood samples were collected immediately after feeding and at 2 and 4 h after feeding. Rumen fluid was taken from the rumen by a stomach tube connected to a vacuum pump. Rumen fluid was tasted for pH and temperature using a portable pH temperature meter (HANNA Instruments HI 8424 microcomputer, Kallang Way, Singapore) and ammonia-nitrogen (NH_3_-N) using a Kjeltech Auto 1030 Analyzer [16]. Rumen fluid was used for direct counts of bacteria and protozoa using Galyean’s methods [19]. Volatile fatty acids (VFA) were measured using high pressure liquid chromatography [20]. Determination of rumen CH_4_ emissions using VFA concentration profiles according to Moss et al. [21] was as follows: CH_4_ emission = 0.45 (acetate) − 0.275 (propionate) + 0.40 (butyrate).

Blood samples of 6 mL were taken from a jugular vessel, kept in Ethylenediaminetetraacetic acid (EDTA) tubes and used for blood urea nitrogen (BUN) analysis according to Crocker [22].

### 2.3. Analysis of Statistic

Statistical analyses were performed using general linear model (GLM) procedure [23]. The results are presented as mean values and standard error of the means. Trend of PS leaf powder levels responded was performed by orthogonal polynomials. Significance was declared at *p* < 0.05 as representing statistically significant differences.

## 3. Results and Discussion

### 3.1. Diet Composition

The concentrates and rice straw contained 13.6% and 3.3% crude protein (CP), respectively. The PS leaf powder consisted of 26.2% DM, 91.1% OM, 8.9% ash, 63.4% NDF and 38.6% ADF (Table 1). Moreover, the PS leaf powder contained 19.1% CP. Thus, it could be an alternative protein source for cattle. However, the may influence the nutritional value of the plant, as previously reported by Kratchanova et al. [24] who determined that PS leaf meal contained 23% and 28% CP when extracted by water and ethanol, respectively. Moreover, PS consisted of high Ca, β-carotein and volatile oil, ligans, alkaloids, flavonoids and polyphenols, which are antioxidants [24]. In the current study, flavonoid content in PS was 91.02 mg/g which was close to that reported by Hafizah et al. [18], who determined that the PS contained a high amount of flavonoid compounds at 95.07 mg/g.

### 3.2. Intake and Digestibility

Feed intake is normally explained in relation to BW^0.75^, the index for general metabolism, or more simply as a percentage of BW. DM intake of rice straw and total intake in terms of g/kg BW^0.75^ differed among treatments and were linearly greatest when PS was added at 2.4 g/head/d (*p* < 0.05) (Table 2). When animals consume more feed, especially with 2.4 g PS supplementation, they can be expected to have a higher substrates supply for animal host. Similar to the finding of Wang et al. [15], piglets supplemented with 50 mg/kg PS leaf powder had the highest increase (33%) of feed intake values due to the flavor and palatability of PS. PS leaf powder did not affect the intake of nutrients and digestibility of OM, CP, NDF or ADF (Table 2). Nevertheless, digestibility of DM was found to increase by 4.8% in cattle when added at 2.4 g/h/d as compared to the other control treatments. This result could be attributed to protozoa, which are capable of ingesting fibrolytic bacteria; therefore, suppressing these protozoa is expected to increased number of fibrolytic bacteria in the rumen [6], which can enhance feed digestibility in animals. However, the present results have only determined total bacterial population but could not confirm whether fibrolytic bacteria enhance of improve feed digestion.

### 3.3. Characteristics of Rumen Fermentation, Bacteria and Protozoa, Blood Urea Nitrogen

Ruminal ecology is shown in Table 3. PS leaf powder did not affect rumen fermentation, suggesting that these ranges of ruminal pH and ruminal temperature were considered an optimal level for microbial activity [25]. The content of NH_3_-N in the rumen is a main source of nitrogen for microorganism protein synthesis [26]. It did not vary among treatments and was similar to results previously studied by Wanapat and Pimpa [27].

The bacterial population was similar across PS levels (*p* > 0.05). However, the protozoal count was lower in animals fed with the supplement at 2.4 g/head/d (*p* < 0.05) (Table 3). The presence of flavonoid content in the PS leaf powder is probably responsible for its antimicrobial activity [28]. Many flavonoids have dependent effects on the protozoal population [29]. This could be because the flavonoid compound may directly affect the protozoa, which inhibits cell wall synthesis or nucleic acid synthesis. This was confirmed by Hernandez et al. [30], who indicated that flavonoids can be used in animal diets to linearly reduce protozoal counts. This result is close to that of Paula et al. [31], who also found that increasing the level of flavonoid compounds can reduce the Entodinium protozoal population in water buffaloes.

BUN concentrations are presented in Table 3. BUN concentrations are highly related with the concentration of NH_3_ in the rumen; therefore, no significant differences were found in BUN concentration. The average value of BUN ranged from 10.86 to 11.76 mg/dL and was close to the value previously revealed by Cherdthong et al. [25]. Similar rumen NH_3_-N contents also resulted in similar BUN contents [4].

The PS leaf powder did not affect TVFA and acetic acid (C2), butyric acid (C4), C2 to propionic acid (C3) and C2 to C3 to C4 (Table 4). However, concentration of C3 at 4 h post feeding and mean value linearly increased with supplementation of PS leaf powder at 2.4 g/d. The enhancement of C3 content might be related to the improvement of feed intake, DM digestion, and microorganism activity [32]. In addition, C3 production available was associated with hydrogen consumption in the rumen, whereas C2 and C4 formation processes released hydrogen. Therefore, a changing of rumen fermentation from C2 to C3 will lower hydrogen release and decrease CH_4_ production [33,34].

### 3.4. Volatile Fatty Acid and CH_4_ Production

Supplementation of PS on CH_4_ estimation in the rumen of beef cattle is shown in Table 4. CH_4_ estimation was altered across PS level supplementation (*p* < 0.05). The 4 h post-feeding and mean values of CH_4_ concentration were linearly reduced with PS supplementation (*p* < 0.05). An increase of PS leaf powder at 2.4 g decreased CH_4_ after feeding 4 h by 21.33% when compared to no PS leaf powder. The reason is probably that the flavonoid compounds present in PS leaf powder suppressed methanogenesis indirectly by linearly reducing the protozoal population, thus reducing methanogenic bacteria symbiotically associated with the protozoa by inhibiting cell wall synthesis or nucleic acid synthesis. Patra and Saxena [9] revealed that flavonoids directly inhibit methanogenic bacteria, and decreased protozoa related to rumen CH_4_ synthesis. In addition, Oskoueian et al. [35] demonstrated that the supplementation of flavonoid-rich plant extracts (PEs) studied in in vitro and in vivo might decrease methanogenic bacteria and CH_4_ production.

## 4. Conclusions

Supplementation of PS leaf powder at 2.4 g/head/d improved feed intake, dry matter digestibility, and propionate concentration, whereas the numbers of protozoa and concentration of CH_4_ were decreased. Thus, supplementation of PS leaf powder in ruminant feeding is recommended for manipulating rumen efficiency. However, future studies on the effects of PS leaf powder should be clarify this through production trial.

## Figures and Tables

**Table 1 animals-09-00130-t001:** The proportion of ingredients in concentrate and nutrient contents of concentrate, rice straw and *Piper sarmentosum* (PS) leaf meal used in experiment.

Item	Concentrate	Rice Straw	PS Leaf Meal
Ingredients, kg DM			
Cassava chip	55.00		
Rice bran	11.0		
Coconut meal	12.90		
Palm kernel meal	13.50		
Urea	2.60		
Pure sulfur	1.00		
Mineral premix	1.00		
Molasses, liquid	2.00		
Salt	1.00		
Chemical composition			
Dry matter, %	91.30	92.10	26.20
Organic matter, %DM	87.00	80.30	91.10
Crude protein, %DM	13.60	3.30	19.10
Neutral detergent fiber, %DM	12.20	65.80	63.40
Acid detergent fiber, %DM	8.40	40.20	38.60
Total flavonoid, mg/g			91.02

**Table 2 animals-09-00130-t002:** Effect of supplementation of *Piper sarmentosum* (PS) leaf meal on feed used and nutrient digestibility in animals.

Item	Concentrations of PS Sup., g/d				SEM	*P*	Contrast	
	0	0.6	1.2	2.4			Lin.	Qua.
DM intake								
Rice straw								
kg/day	2.25	2.32	2.36	2.45	0.11	0.81	ns	ns
g/kg BW^0.75^	52.76 ^a^	54.13 ^ab^	55.06 ^ab^	57.45 ^b^	1.22	0.03	0.02	ns
Concentrate								
kg/day	0.75	0.75	0.75	0.75	0.23	0.71	ns	ns
g/kg BW^0.75^	17.47	17.50	17.50	17.47	0.71	0.55	ns	ns
*Piper sarmentosum* leaf meal								
g/day	0.00	0.60	1.20	2.40	-	-	-	-
g/kg BW^0.75^	0.000	0.014	0.028	0.056	-	-	-	-
Total intake								
kg/day	3.00	3.07	3.11	3.20	0.56	0.34	ns	ns
g/kg BW^0.75^	70.23 ^a^	71.64 ^a^	72.59 ^ab^	74.97 ^b^	1.13	0.02	0.03	ns
Nutrient intake, kg								
Dry matter	3.00	3.07	3.11	3.20	0.96	0.34	ns	ns
Organic matter	2.70	2.77	2.80	2.88	0.76	0.26	ns	ns
Crude protein	0.53	0.55	0.55	0.57	0.09	0.18	ns	ns
Neutral detergent fiber	1.52	1.56	1.59	1.65	0.634	0.33	ns	ns
Acid detergent fiber	0.93	0.96	0.91	1.01	0.18	0.76	ns	ns
Digestibility								
Dry matter, %	64.05 ^a^	65.33 ^a^	66.52 ^ab^	68.85 ^b^	1.02	0.03	0.04	ns
Organic matter, %DM	72.64	73.03	73.54	75.83	2.50	0.84	ns	ns
Crude protein, %DM	54.05	55.34	55.95	57.88	1.70	0.71	ns	ns
Neutral detergent fiber, %DM	50.36	51.92	52.00	53.17	1.63	0.55	ns	ns
Acid detergent fiber, %DM	34.23	35.64	35.35	36.51	0.98	0.44	ns	ns

^a,b^ Values within the same row not bearing a common superscript differ (*p* < 0.05); ns = non-significant; Sup. = Supplementation; Qua. = Quadratic; Lin. = Linear.

**Table 3 animals-09-00130-t003:** Effect of supplementation of *Piper sarmentosum* (PS) leaf meal on ruminal fermentation, rumen microorganisms and blood metabolite.

Item	Concentrations of PS Sup., g/d				SEM	*P*	Contrast	
	0	0.6	1.2	2.4			Lin.	Qua.
Rumen ecology								
Ruminal pH								
0 h after feeding	7.13	7.01	7.18	7.06	0.10	0.66	ns	ns
4 h after feeding	6.97	6.99	7.02	6.9	0.06	0.61	ns	ns
Mean	7.05	7.00	7.10	6.98	0.06	0.58	ns	ns
Ruminal temperature, °C								
0 h after feeding	39.46	39.45	39.71	38.60	0.95	0.74	ns	ns
4 h after feeding	40.70	40.26	40.21	39.96	0.68	0.50	ns	ns
Mean	40.08	39.86	39.96	39.28	0.49	0.68	ns	ns
NH_3_-N concentration, mg/dL								
0 h after feeding	13.00	13.25	13.00	14.05	0.17	0.51	ns	ns
4 h after feeding	14.50	14.25	15.50	15.50	0.13	0.52	ns	ns
Mean	13.75	13.75	14.25	14.78	0.11	0.49	ns	ns
Blood urea nitrogen, mg/dl								
0 h after feeding	10.51	11.01	10.98	11.21	1.19	0.41	ns	ns
4 h after feeding	11.21	12.11	11.25	12.31	1.47	0.42	ns	ns
Mean	10.86	11.56	11.12	11.76	1.23	0.37	ns	ns
Ruminal microbes, cell/mL								
Bacteria, ×10^9^								
0 h after feeding	1.00	1.08	1.07	1.04	0.53	0.71	ns	ns
4 h after feeding	1.17	1.15	1.09	1.05	0.45	0.41	ns	ns
Mean	1.09	1.12	1.08	1.05	0.40	0.97	ns	ns
Protozoa, ×10^6^								
0 h after feeding	9.05 ^a^	5.00 ^b^	6.26 ^b^	3.12 ^c^	1.30	0.03	0.01	ns
4 h after feeding	9.25 ^a^	6.25 ^b^	6.37 ^b^	3.50 ^c^	1.67	0.04	0.04	ns
Mean	9.15 ^a^	5.63 ^b^	6.32 ^b^	3.31 ^c^	1.54	0.01	0.02	ns

^a–c^ Values within the same row not bearing a common superscript differ (*p* < 0.05); ns = non-significant; Sup. = Supplementation; Qua. = Quadratic; Lin. = Linear.

**Table 4 animals-09-00130-t004:** Effect of supplementation of *Piper sarmentosum* (PS) leaf meal on ruminal volatile fatty acid (VFA) profile and methane (CH_4_) prediction.

Item	Concentrations of PS Sup., g/d				SEM	*P*	Contrast	
	0	0.6	1.2	2.4			Lin.	Qua.
Total VFA, mmol/L								
0 h after feeding	95.66	93.23	93.01	92.11	29.93	0.49	ns	ns
4 h after feeding	102.60	100.20	100.10	98.70	30.93	0.95	ns	ns
Mean	99.13	96.72	96.56	95.41	28.79	0.76	ns	ns
VFA profiles, mol/100 mol								
Acetic acid								
0 h after feeding	69.01	70.20	69.99	68.29	1.84	0.87	ns	ns
4 h after feeding	71.69	72.39	71.99	70.09	3.02	0.20	ns	ns
Mean	70.35	71.30	70.99	69.19	2.34	0.31	ns	ns
Propionic acid								
0 h after feeding	18.02	19.89	19.91	20.11	1.48	0.24	ns	ns
4 h after feeding	19.91 ^a^	20.21 ^a^	22.45 ^b^	24.89 ^c^	0.59	0.03	0.04	ns
Mean	18.97 ^a^	20.05 ^a^	21.18 ^b^	22.50 ^c^	0.22	0.02	0.03	ns
Butyric acid								
0 h after feeding	12.97	9.91	10.10	11.60	3.60	0.33	ns	ns
4 h after feeding	8.40	7.40	5.56	5.02	2.38	0.18	ns	ns
Mean	10.69	8.66	7.83	8.31	1.96	0.67	ns	ns
Acetic/propionic acid ratio								
0 h after feeding	3.83	3.53	3.52	3.40	0.32	0.33	ns	ns
4 h after feeding	3.60	3.58	3.21	2.82	0.24	0.28	ns	ns
Mean	3.71	3.56	3.35	3.08	0.12	0.33	ns	ns
Acetic plus butyric/propionic ratio								
0 h after feeding	4.55	4.03	4.02	3.97	0.52	0.40	ns	ns
4 h after feeding	4.02	3.95	3.45	3.02	0.33	0.21	ns	ns
Mean	4.27	3.99	3.72	3.44	0.44	0.39	ns	ns
CH_4_ estimation, mM/L								
0 h after feeding	32.25	31.12	29.78	29.91	1.87	0.51	ns	ns
4 h after feeding	30.85 ^a^	28.69 ^b^	27.14 ^b^	24.45 ^c^	0.51	0.02	0.03	ns
Mean	31.55 ^a^	29.91 ^b^	28.46 ^b^	27.18 ^c^	0.98	0.04	0.02	ns

^a–c^ Values within the same row not bearing a common superscript differ (*p* < 0.05); ns = non-significant; Sup. = Supplementation; Qua. = Quadratic; Lin. = Linear.

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
