# Peer review of "Effects of Supplementation of *Piper sarmentosum* Leaf Powder on Feed Efficiency, Rumen Ecology and Rumen Protozoal Concentration in Thai Native Beef Cattle"

_animals, 2019, doi:10.3390/ani9040130_

Round 1
Reviewer 1 Report
I cannot agree with you. Please answer to my questions.
L36; However, future studies should be conducted under the production field. --- I think this is not necessary in Abstracts.
L56; positive effects of flavonoid-rich plants on CH4 emission --- What are positive effects ? Simply it can be understood that CH4 emission will increase.
L108; [19].Volatile --- insert space
L118; P<0.05 --- insert space ?
L118; Significance was declared at P<0.05 as representing statistically significant differences. --- Are you following this rule? P < 0.05
L122; 8.9% OM, --- Is this all right ?
L129; Thus, PS containing flavonoids could be manipulated for rumen fermentation and mitigating CH4 production. --- Why can you say so here ? I feel sudden without any reasons.
L134; P<0.05 --- insert space ?
L137; digestibility of nutrients (DM, OM, CP; P<0.08) tended to be high in cattle added at 2.4 g/h/d as compared to the other treatments --- P-values are very high, DM 0.93, OM 0.84, CP0.71. I think you cannot say so.
L139; This result could be attributed to linearly reduced number of protozoa --- Why do digestibility increase by reducing number of protozoa ? Do you think increasing number of protozoa give negative effects for ruminants ?
L139; increased number of fibrolytic bacteria --- Can you say so in this study ?
L161; Concentrations --- concentrations
BUN Concentrations are highly related with the concentrations of NH3 in the rumen. --- How about in this study ?
L177; P<0.05 --- insert space ?
L190; could improve dry matter digestibility --- P-value is 0.93. Can you say so ?
SEM is 3.32. Are 65.05 in 0, 65.33 in 0.6, 66.52 in 1.2 and 68.85 in 2.4 significantly different ? I cannot believe.
L192; supplementation of PS leaf powder in ruminant feeding is recommended. --- Can you say so ? I think PS leaf powder has only one effect (reducing protozoa) for ruminant.
L193; practical animal --- What do you mean ? Were these experimental animals not practical ? In special situation ?
Table 2.
Concentrate intake significantly reduced in 0.6 and 1.2 compared to 0. How do you think this ?
Total intake in 0 is 2.96kg/day. 2.30 rice straw + 0.81 concentrate =3.11. Why are they different ?
Total intake in 0.6 is 3.00kg/day. 2.32 rice straw + 0.74 concentrate =3.06. Why are they different ?
Total intake in 1.2 is 3.14kg/day. 2.36 rice straw + 0.64 concentrate =3.00. Why are they different ?
Total intake in 2.4 is 3.22kg/day. 2.45 rice straw + 0.82 concentrate =3.27. Why are they different ?
Total intake kg/day (2.96, 3.00, 3.14, 3.22) and Nutrient intake Dry matter, kg (2.90, 3.00, 3.10, 3.20) --- Why are they different ?
ns = non-significant --- Why do you show 0.07, 0.06, 0.08. You say in L118 (Significance was declared at P<0.05). It is inconsistent.
Table 3.
Blood urea nitrogen, Mean --- P-value is 0.37. I think these data are not significantly different.
Protozoa, Mean ---
Protozoa, Mean in 0 is 9.38. (9.05 0h + 9.25 4h) /2 = 9.15. Why are they different ?
Protozoa, Mean in 0.6 is 5.65. (5.00 0h + 6.25 4h) /2 = 5.63. Why are they different ?
Protozoa, Mean in 1.2 is 6.52. (6.26 0h + 6.37 4h) /2 = 6.32. Why are they different ?
Protozoa, Mean in 2.4 is 3.33. (3.12 0h + 3.50 4h) /2 = 3.31. Why are they different ?
ns = non-significant --- Why do you show 0.09, 0.06, 0.07. You say in L118 (Significance was declared at P<0.05). It is inconsistent.
Table 4.
Propionic acid, 0h and 4h, all ns.
Why only is P-value 0.02 in Mean ? I cannot understand this. I think 4h after feeding is very important for evaluating feed additives. Therefore I think PS did never affect Propionic acid.
CH4 estimation, 0h and 4h, all ns.
Why only is P-value 0.04 in Mean ? I cannot understand this. I think 4h after feeding is very important for evaluating feed additives. Therefore I think PS did never affect CH4 estimation.
Author Response
Response to Reviewer #1:
I cannot agree with you. Please answer to my questions.
Response: Thanks and we have tried our best in order to improve our manuscript following your comments. Please see more detail in the manuscript.
L36; However, future studies should be conducted under the production field. --- I think this is not necessary in Abstracts.
Response: We have removed. Please see in manuscript.
L56; positive effects of flavonoid-rich plants on CH4 emission --- What are positive effects ? Simply it can be understood that CH4 emission will increase.
Response: We have changed to “….In addition, the negative effects of flavonoid-rich plants….”
L108; [19].Volatile --- insert space
Response: We have modified. Please see in manuscript.
L118; P<0.05 --- insert space ?
Response: We have modified. Please see in manuscript.
L118; Significance was declared at P<0.05 as representing statistically significant differences. --- Are you following this rule? P < 0.05
Response: Yes, we are and now we have modified already.
L122; 8.9% OM, --- Is this all right ?
Response: Thanks, we have changed OM content to 91.1% DM and 8.9% ash.
L129; Thus, PS containing flavonoids could be manipulated for rumen fermentation and mitigating CH4 production. --- Why can you say so here ? I feel sudden without any reasons.
Response: We have removed. Please see in manuscript.
L134; P<0.05 --- insert space ?
Response: We have modified. Please see in manuscript.
L137; digestibility of nutrients (DM, OM, CP; P<0.08) tended to be high in cattle added at 2.4 g/h/d as compared to the other treatments --- P-values are very high, DM 0.93, OM 0.84, CP0.71. I think you cannot say so.
Response: We have re-analyzed statistic and now revised text and table already. Please see in manuscript.
L139; This result could be attributed to linearly reduced number of protozoa --- Why do digestibility increase by reducing number of protozoa ? Do you think increasing number of protozoa give negative effects for ruminants ?
Response: revised as “This result could be attributed to protozoa, which are capable of ingesting fibrolytic bacteria; therefore, suppressing these protozoa is expected to increased number of fibrolytic bacteria in the rumen [6], which can enhance feed digestibility in animals.” Please see in manuscript.
L139; increased number of fibrolytic bacteria --- Can you say so in this study ?
Response: Even though fibrolytic bacteria population did not determined in this study, we would like to provide possibility reason to support why DM digestibility increased. However, this sentence has been modified as mention above. Please see in manuscript.
L161; Concentrations --- concentrations
Response: We have modified. Please see in manuscript.
BUN Concentrations are highly related with the concentrations of NH3 in the rumen. --- How about in this study ?
Response: Sorry, after we have recalculated and found that BUN not differ among treatments. Now, we already updated and revised this sentence as “BUN concentrations are presented in Table 3. BUN concentrations are highly related with the concentration of NH3 in the rumen; therefore, no significant differences were found in BUN concentration.” Please see in manuscript.
L177; P<0.05 --- insert space ?
Response: We have modified. Please see in manuscript.
L190; could improve dry matter digestibility --- P-value is 0.93. Can you say so ?
SEM is 3.32. Are 65.05 in 0, 65.33 in 0.6, 66.52 in 1.2 and 68.85 in 2.4 significantly different ? I cannot believe.
Response: We have re-analyzed statistic and now revised text and table already. Please see in manuscript.
L192; supplementation of PS leaf powder in ruminant feeding is recommended. --- Can you say so ? I think PS leaf powder has only one effect (reducing protozoa) for ruminant.
Response: We have already modified as “Supplementation of PS leaf powder at 2.4 g/h/d improved feed intake, dry matter digestibility, and propionate concentration, whereas the numbers of protozoa and concentration of CH4 were decreased. Thus, supplementation of PS leaf powder in ruminant feeding is recommended for manipulating rumen efficiency. However, future studies on the effects of PS leaf powder should be clarify this through production trial.” Please see in the text.
L193; practical animal --- What do you mean ? Were these experimental animals not practical ? In special situation ?
Response: It mean “production trial” eg dairy cow, fattening beef in order to see an effect on animal performances efficiency (milk, meat). Thus, we have indicated in conclusion already.
Table 2.
Concentrate intake significantly reduced in 0.6 and 1.2 compared to 0. How do you think this ?
Response: We have re-check data and re-analyzed statistic and then now revised text and table already. Please see in manuscript.
Total intake in 0 is 2.96kg/day. 2.30 rice straw + 0.81 concentrate =3.11. Why are they different ?
Response: We have re-check data and re-analyzed statistic and then now revised text and table already. Please see in manuscript.
Total intake in 0.6 is 3.00kg/day. 2.32 rice straw + 0.74 concentrate =3.06. Why are they different ?
Response: We have re-check data and re-analyzed statistic and then now revised text and table already. Please see in manuscript.
Total intake in 1.2 is 3.14kg/day. 2.36 rice straw + 0.64 concentrate =3.00. Why are they different ?
Response: We have re-check data and re-analyzed statistic and then now revised text and table already. Please see in manuscript.
Total intake in 2.4 is 3.22kg/day. 2.45 rice straw + 0.82 concentrate =3.27. Why are they different ?
Response: We have re-check data and re-analyzed statistic and then now revised text and table already. Please see in manuscript.
Total intake kg/day (2.96, 3.00, 3.14, 3.22) and Nutrient intake Dry matter, kg (2.90, 3.00, 3.10, 3.20) --- Why are they different ?
Response: We have re-check data and re-analyzed statistic and then now revised text and table already. Please see in manuscript.
ns = non-significant --- Why do you show 0.07, 0.06, 0.08. You say in L118 (Significance was declared at P<0.05). It is inconsistent.
Response: We have re-check data and re-analyzed statistic and then now revised text and table already. Please see in manuscript.
Table 3.
Blood urea nitrogen, Mean --- P-value is 0.37. I think these data are not significantly different.
Response: We have re-check data and re-analyzed statistic and then now revised text and table already. Please see in manuscript.
Protozoa, Mean ---
Protozoa, Mean in 0 is 9.38. (9.05 0h + 9.25 4h) /2 = 9.15. Why are they different ?
Response: We have re-check data and re-analyzed statistic and then now revised text and table already. Please see in manuscript.
Protozoa, Mean in 0.6 is 5.65. (5.00 0h + 6.25 4h) /2 = 5.63. Why are they different ?
Response: We have re-check data and re-analyzed statistic and then now revised text and table already. Please see in manuscript.
Protozoa, Mean in 1.2 is 6.52. (6.26 0h + 6.37 4h) /2 = 6.32. Why are they different ?
Response: We have re-check data and re-analyzed statistic and then now revised text and table already. Please see in manuscript.
Protozoa, Mean in 2.4 is 3.33. (3.12 0h + 3.50 4h) /2 = 3.31. Why are they different ?
Response: We have re-check data and re-analyzed statistic and then now revised text and table already. Please see in manuscript.
ns = non-significant --- Why do you show 0.09, 0.06, 0.07. You say in L118 (Significance was declared at P<0.05). It is inconsistent.
Response: We have now revised text and table already. Please see in manuscript.
Table 4.
Propionic acid, 0h and 4h, all ns.
Why only is P-value 0.02 in Mean ? I cannot understand this. I think 4h after feeding is very important for evaluating feed additives. Therefore I think PS did never affect Propionic acid.
Response: We have re-check data and re-analyzed statistic and then now revised text and table already. Please see in manuscript.
CH4 estimation, 0h and 4h, all ns.
Why only is P-value 0.04 in Mean ? I cannot understand this. I think 4h after feeding is very important for evaluating feed additives. Therefore I think PS did never affect CH4 estimation.
Response: We have re-check data and re-analyzed statistic and then now revised text and table already. Please see in manuscript.

Reviewer 2 Report
This work needs to be edited for English. English is poor. There are problems with almost every sentence. Please have an English reviewer/native speaker help with the manuscript.
Author Response
This work needs to be edited for English. English is poor. There are problems with almost every sentence. Please have an English reviewer/native speaker help with the manuscript.
Response: We had revised about all of the points raised by the reviewers and some points that we found that they should be revised. The revised paper was proved by the native English speaker. We submitted our Manuscript to the company namely Papercheck (https://www.papercheck.com/) located in California, United States for English grammar edited already (Order no. 751820). Therefore, throughout manuscript have been improved in English language and indicated with red color highlight.
Thank you very much!

Round 2
Reviewer 1 Report
Although suspicious changes statistical analysis claims about, I have no choice but to believe.
Author Response
Although suspicious changes statistical analysis claims about, I have no choice but to believe. Response: Thank you very much for your believed our re-calculation of this research.
Reviewer 2 Report
Thank you for addressing my concerns regarding English. Please check, in a couple of places you still say PS leaves as opposed to PS leaf meal.
Please correct your stats section as now you are doing orthogonal comparisons, but you still say Duncan's Multiple range test.
Author Response
Thank you for addressing my concerns regarding English. Please check, in a couple of places you still say PS leaves as opposed to PS leaf meal.
Response: We have already changed all from “PS leaves powder” to “PS leaf meal”. Please see in Lines eg 82, 94, 96, 131 and in Table 1, 2, 3, 4. Please see in the manuscript.
Please correct your stats section as now you are doing orthogonal comparisons, but you still say Duncan's Multiple range test.
Response: We have removed “Duncan's Multiple range test” from the manuscript. Please see in the section of Statistic analysis.
Thank you very much!
This manuscript is a resubmission of an earlier submission. The following is a list of the peer review reports and author responses from that submission.
Round 1
Reviewer 1 Report
I cannot understand your study because you do not show the statistical difference especially between 0 and 2.4.
L104; Means were compared using Duncan's New Multiple Range Test. --- Please show in Table by using a,b,c in superscripts.
Table 1. PS leaves powder, Organic matter, %DM, 8.9 --- Is this OK ? Ash ?
Table 2. Why you did not compare among control, 0.6, 1.2 and 2.4 ?
You show at Item Concentrations of PS supplementation, g/d --- Is this OK ? kg/d ?
Control, Nutrient intake, Dry matter, 2.90 --- Why did you use superscript ?
Table 3. Why you did not compare among control, 0.6, 1.2 and 2.4 ?
You show at Item Concentrations of PS supplementation, g/d --- Is this OK ? kg/d ?
Table 4. Why you did not compare among control, 0.6, 1.2 and 2.4 ?
You show at Item Concentrations of PS supplementation, g/d --- Is this OK ? kg/d ?
L211, L270; Why you use underline ?
Reviewer 2 Report
I recommend that you work with someone proficient in English to improve the paper. The results are very interesting. However, I think the statistical analyses need to be rerun. Your study is designed to test orthogonal comparisons, but you evaluated multi-range tests. You used SAS. I suggest that you rerun the data and evaluate linear, quadratic and cubic effects. In fact, when evaluating the data, it appears that linear and quadratic effects may be occurring.
On lines 125-128- you cannot say that, the data are not even approaching significance.
Please redo the statistical analyses and have someone proficient in English review the manuscript.
Reviewer 3 Report
The manuscript brings interesting information about the use of vegetables alternatives to improve rumen ecology and performance; I missed some serum metabolite analyses to assess the energetic (BHB, NEFA, etc.) and protein profiles (BUN, total proteins, albumin).
Furthermore, I have some comments and questions:
Abstract
L 22. The sentence “The BUN concentration tended with increasing PS leaf powder” is incomplete. In addition, I don’t see the tendency because the P value is almost 0.4.
Material and methods
L 64-65. It would be interesting to know the age of the beef cattle used.
L 86-88. The information is not clear, please rewrite it.
L 93. The authors don’t describe the method use to obtain the rumen fluid, and it’s very interesting know if was by rumenocentesis or via esophageal tubing, especially when the pH value is determined.
L 100-101. Why the authors don’t determine more metabolic parameters to assess the energetic profile, as BHB and NEFA? It would be very interesting and could reinforce the results obtained
Results and discussion
L 110. The percentage of NDF (70.6 %) don’t match with the value showed in table-1 (63.4%)
L 111. The CP value is not showed in table-1.
L 116-117. The content of flavonoid in PS is different in line 116 than in table-1. In table-1 the content is 91.02 mg/g, the same than in the Hafizah et al. study.
L 146. When the authors used the word “tended” the readers could understand that the significance level was between 0.05-0.1, and it’s not the case. Please rewrite the sentence.
L 153. The volatile fatty acid profile is described in table-4 not in table-3. Please change it in the MS.
L 163-164. The increased is only significant at 2.4 g/d supplementation, when a 0.6 g/d dose is added there is even an increased in CH4 levels. Is there any explanation
Conclusion
The authors should highlight more the findings, I miss the positive implications that the supplement could have. At least, in the abstract, the authors postulate the PS leaf powder as an alternative feeding regime. Please improve the conclusions.
Tables
All tables: When presenting data we follow the general rule of three significant figures. On my opinion, number of decimals depends on the size of the number, and in total 3 significant figures seems adequate.
Table-2: The DM intake of PS is expressed as kg/day; in material and methods sections is g/day. Please standardize it.